# Ultrastructural study of sperm cells in Acanthocolpidae: the case of *Stephanostomum murielae* and *Stephanostomoides tenuis* (Digenea)

Abdoulaye J.S. Bakhoum[1,2], Yann Quilichini[1], Jean-Lou Justine[3], Rodney A. Bray[4], Cheikh T. Bâ[2] and Bernard Marchand[1]

[1] SERME Service d'Étude et de Recherche en Microscopie Electronique, UMR 6134-SPE CNRS—Università di Corsica, Corte, Corsica, France
[2] Laboratory of Evolutionary Biology, Ecology and Management of Ecosystems, Cheikh Anta Diop University of Dakar, Dakar, Senegal
[3] ISYEB, Institut de Systématique, Évolution, Biodiversité (UMR7205 CNRS, EPHE, MNHN, UPMC), Muséum National d'Histoire Naturelle, Paris Cedex, France
[4] Department of Life Sciences, Natural History Museum, London, United Kingdom

Corresponding author
Yann Quilichini,
quilichini@univ-corse.fr

## ABSTRACT

The mature spermatozoa of *Stephanostomum murielae* and *Stephanostomoides tenuis* are described by transmission electron microscopy. They present several ultrastructural features previously reported in other digeneans. Their spermatozoa possess two axonemes of different length showing the 9 + '1' trepaxonematan pattern, four attachment zones, two mitochondria (with an anterior moniliform one in *S. murielae*), a nucleus, two bundles of parallel cortical microtubules, external ornamentation of the plasma membrane, spine-like bodies and granules of glycogen. The main differences between the mature spermatozoon of *S. murielae* and *S. tenuis* are the maximum number of cortical microtubules, the morphology of the anterior spermatozoon extremity and the anterior mitochondrion. This study is the first concerning members of the family Acanthocolpidae. The main ultrastructural characteristics discussed are the morphology of the anterior and posterior spermatozoon extremities, antero-lateral electron dense material, external ornamentations, spine-like bodies and number and morphology of mitochondria. In addition, the phylogenetic significance of all these ultrastructural features is discussed and compared to molecular results in order to highlight the complex relationships in the Digenea.

## INTRODUCTION

The Platyhelminthes are invertebrate organisms characterized by the absence of fossils. Therefore, to understand the relationships between the species, research can be carried out only on extant taxa. Among the numerous approach possibilities, we have chosen

to study the cells which present a great variabiliy among the animals in general and the platyhelminthes in particular: the sperm cells.

Members of the Acanthocolpidae Lühe, 1906 (Digenea, Platyhelminthes) are parasites of marine teleost fishes and occasionally of sea snakes. They are mainly characterized by a spinous tegument, the lack of an external seminal vesicle and the presence of an uterine seminal receptacle (*Bray, 2005a*). *Lühe (1909)* first erected this clade to the family rank. Since then, this family has accumulated several genera and tends to become a so-called "catch-all group" because many of the genera were included purely for convenience of identification (*Bray, 2005a*; *Bray et al., 2009*). Morphological and/or molecular studies have been carried out in order to highlight the complex relationships within the Acanthocolpidae and also the place of this family in the Digenea system (*Cribb et al., 2001*; *Olson et al., 2003*; *Bray, 2005b*; *Bray et al., 2009*). In fact, *Bray (2005b)* in *Jones, Bray & Gibson (2005)* morphological key to the Trematoda, included ten families in the superfamily Lepocreadioidea Odhner, 1905. Those are the Acanthocolpidae Lühe 1906; Apocreadiidae Skrjabin, 1942; Brachycladiidae Odhner, 1905; Deropristidae Cable and Hunninen, 1942; Enenteridae Yamaguti, 1958; Gorgocephalidae Manter, 1966; Gyliauchenidae Fukui, 1929; Lepocreadiidae Odhner, 1905; Liliatrematidae Gubanov, 1953 and Megaperidae Manter, 1934. According to *Bray & Cribb*'s (*2012*) recent re-organisation of the Lepocreadioidea, three of these families, namely the Acanthocolpidae, Apocreadiidae and Brachycladiidae, should be excluded from the superfamily and placed elsewhere in the digenean system. *Pulis et al. (2014)*, utilising molecular means, found that the Megaperidae should be subsumed in the family Apocreadiidae. Two other families (Deropristidae and Liliatrematidae) wait for supplementary studies. Finally, the remaining families, including two new families Aephnidiogenidae and Lepidapedidae previously considered as subfamilies (*Bray, 2005c*), form a monophyletic group now known under the name Lepocreadioidea.

Ultrastructural studies of spermatozoa have been carried out on seven species belonging to five of the families mentioned above. Those are the aephnidiogenid *Holorchis micracanthum* (*Bâ et al., 2011*), the apocreadiid *Neoapocreadium chabaudi* (*Kacem et al., 2010*), the lepocreadiids *Hypocreadium caputvadum* (*Kacem et al., 2012*) and *Opechona bacillaris* (*Ndiaye et al., 2015*), the deropristid *Deropristis inflata* (*Foata, Quilichini & Marchand, 2007*) and the gyliauchenids *Gyliauchen* sp. (*Quilichini et al., 2011*) and *Robphildollfusium fratum* (*Bakhoum et al., 2012*). In some Neodermata such as Cestoda (*Justine, 1991a*; *Justine, 1995*; *Justine, 1998*; *Levron et al., 2010*; *Marigo, 2011*) and Monogenea (*Justine, 1991b*; *Justine, 1995*; *Justine, 1998*; *Marigo, 2011*) several ultrastructural characteristics have been used as valuable tools for phylogenetic inference. Despite the poor ultrastructural data available (about 1%), several studies of Digenea have commented on the usefulness of these characters in understanding their systematic and phylogenetic relationships (*Justine, 1991a*; *Justine, 1995*; *Justine, 2003*; *Euzet, Świderski & Mokhtar-Maamouri, 1981*; *Levron, Ternengo & Marchand, 2003*; *Miquel et al., 2006*; *Quilichini et al., 2010b*; *Bakhoum, 2012*).

The present study presents, for the first time, the spermatological characteristics in two acanthocolpid genera namely *Stephanostomum* and *Stephanostomoides*. Moreover, spermatological features of *Stephanostomum murielae* and *Stephanostomoides tenuis* are compared to those described in other digenean species, particularly lepocreadioideans. In addition, we discuss some principal ultrastructural criteria that could be used in digenean phylogeny and compare our results with those carried out in molecular studies.

## MATERIALS AND METHODS

Live adult specimens of *Stephanostomum murielae* (*Bray & Justine, 2011*) and *Stephanostomoides tenuis* (Manter, 1963) were collected respectively from *Carangoides hedlandensis* (Whitley, 1934) and *Chirocentrus dorab* (Forsskål, 1775) both fish caught off Nouméa, New Caledonia; specimens of both digenean species are kept in the collections of the Muséum National d'Histoire Naturelle, Paris, and Natural History Museum, London (*Bray & Justine, 2011*; *Bray & Justine, 2012*). They were fixed in cold (4 °C) 2.5% glutaraldehyde in 0.1 M sodium cacodylate buffer at pH 7.2, rinsed in 0.1 M sodium cacodylate buffer at pH 7.2, post-fixed in cold (4 °C) 1% osmium tetroxide in the same buffer for 1 h, dehydrated in ethanol and propylene oxide series, embedded in Spurr resin and polymerized at 60 °C for 24 h. Ultrathin sections (60–90 nm) of the seminal vesicle were cut on an ultramicrotome (Power tome PC; RMC Boeckeler®). Sections were placed on 300 and 200 mesh copper grids and stained with uranyl acetate and lead citrate according to *Reynolds*'s (*1963*) methodology. The *Thiéry (1967)* technique was also used to locate glycogen in sections placed on gold grids. Finally, all grids were examined on a Hitachi H-7650 transmission electron microscope (Hitachi H7650; Hitachi, Chiyoda, Tokyo, Japan), operating at an accelerating voltage of 80 kV, in the "Service d'Étude et de Recherche en Microscopie Électronique" of the University of Corsica (Corte, France).

## RESULTS

The interpretation of several cross- and longitudinal sections of the mature spermatozoon of *Stephanostomum murielae* (Figs. 1–3) and *Stephanostomoides tenuis* (Figs. 4–6) allows us to establish four distinctive regions for each species, from the anterior to the posterior spermatozoon extremity (Figs. 1–6).

Region I corresponds to the anterior extremity of the spermatozoon. Both longitudinal and cross-sections in the anterior spermatozoon tip of *Stephanostomum murielae* show one axoneme and few cortical microtubules (about two) (Figs. 1A, 1B and 3I). In *S. tenuis* the anterior spermatozoon tip exhibits both centrioles, corresponding to the axonemes, surrounded by a layer of cortical microtubules (about 25) (Figs. 4A, 4B and 6I). In *S. murielae*, cross-sections in the most posterior areas of the anterior region exhibit an incomplete second axoneme and a continuous layer of about 30 cortical microtubules (Figs. 1C and 3I). When both axonemes are formed, the layer of cortical microtubules becomes discontinuous and their number decreases to about 27 in *S. murielae*, whereas in *S. tenuis* the number of cortical microtubules decreases from 31 to about 23 (Figs. 1D, 4C, 4D, 3I and 6I).

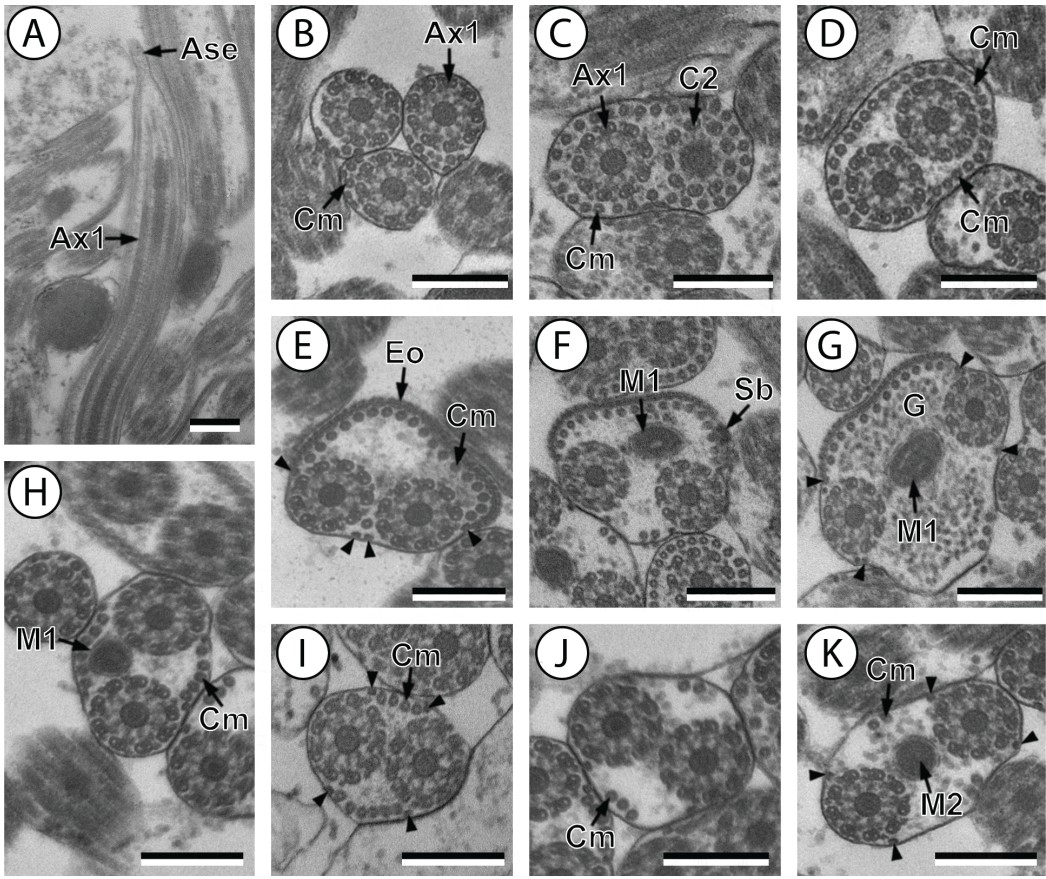

**Figure 1 Transmission electron micrographs of mature spermatozoon of *Stephanostomum murielae* in region I–IV.** (A and B) longitudinal and cross-section sections of region I showing the anterior spermatozoon extremity; (C and D) consecutive cross-sections showing (C) the formation of the second axoneme, continuous layer of cortical microtubules and (D) both axonemes formed accompanied with discontinuous layer of cortical microtubules; (E–G) consecutive cross-sections of region II containing the external ornamentation of the plasma membrane and the first mitochondrion (F and G). Note the presence of spine-like body (F) and four attachment zones (arrowheads); (H–J): region III or transitional areas showing the posterior part of the first mitochondrion, the axonemes and a decreasing number of cortical microtubules; (K): proximal part of region IV showing the second mitochondrion. Scale bars: 0.3 μm. Ase, anterior spermatozoon extremity; Ax1, first axoneme; C2, centriole of the second axoneme; Cm, cortical microtubules; Eo, external ornamentation of the plasma membrane; G, granules of glycogen; M1, first mitochondrion; M2, second mitochondrion; Sb, spine-like body.

Region II is characterized, for each species, by the presence of external ornamentation of the plasma membrane associated with cortical microtubules and the first mito-chondrion. Their maximum number of cortical microtubules is reduced progressively from 19 to 15 in *S. murielae* (Figs. 1E–1G and 3II); in *S. tenuis* the number of cortical microtubules decreases from 19 to 11 (Figs. 4E, 4F, 4H and 6II). It is interesting to note that, unlike *S. murielae*, a first mitochondrion of the *S. tenuis* spermatozoon is moniliform, showing bulges and cords in the ornamented area (Figs. 4F, 4G and 6II). However, in both *S. murielae* and *S. tenuis* spine-like bodies are observed in the ornamented region

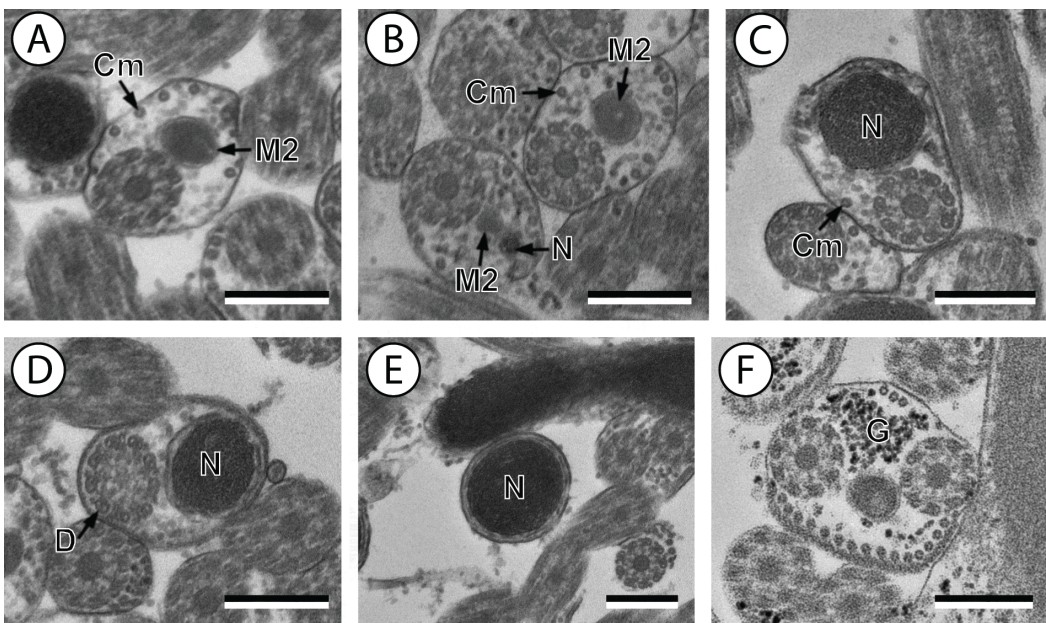

**Figure 2 Transmission electron micrographs of mature spermatozoon of *Stephanostomum murielae* in region IV and V.** (A and B) Cross-section in (A) proximal area of region IV showing second mitochondrion and (B) simultaneous presence of the nucleus and the second mitochondrion; (C) cross-sections showing the second axoneme, the nucleus and few cortical microtubules; (D and E) cross-sections showing disorganization of the second axoneme (D) and a nucleus in posterior tip of the spermatozoon (E); (F) granules of glycogen evidenced according to the Thiéry's test. Scale bars: 0.3 μm. Cm, cortical microtubules; D, doublets; G, granules of glycogen; M2, second mitochondrion; N, nucleus.

(Figs. 1F, 4F, 3II and 6II). Granules of glycogen are also present in this region and as evidenced by Thiéry's test on *S. murielae* material (Fig. 2F).

Region III represents the transitional area before the appearance of the second mitochondrion and nucleus. Passing from its proximal region towards the distal region, the spermatozoon of *S. murielae* presents, in sequence, the first mitochondrion, both axonemes and the two bundles of parallel cortical microtubules of which maximum number is reduced from 11 to about 5 (Figs. 1H–1J and 3III). In *S. tenuis* region III shows no mitochondrion, the two parallel bundles of cortical microtubules and the maximum number of cortical microtubules which decrease from 9 to 6 (Figs. 4I, 4J and 6III). Moreover, the posterior part of region III in *S. tenuis* exhibits in cross-section only two cortical microtubules (Fig. 5A).

Region IV is mainly characterized by the presence of the second mitochondrion in its anterior part and the nucleus in its posterior part. Cross-sections in the anterior part of this region show both axonemes, the second mitochondrion and few parallel cortical microtubules: about 4 in *S. murielae* and 6 in *S. tenuis* (Figs. 1K, 5B, 3IV and 6IV). In the more distal part, one of the axonemes disappears before the appearance of the nucleus. Consequently, in several cross-sections, only one axoneme accompanied by the second mitochondrion is observed (Figs. 2A, 2B and 5C). When the nucleus appears, it

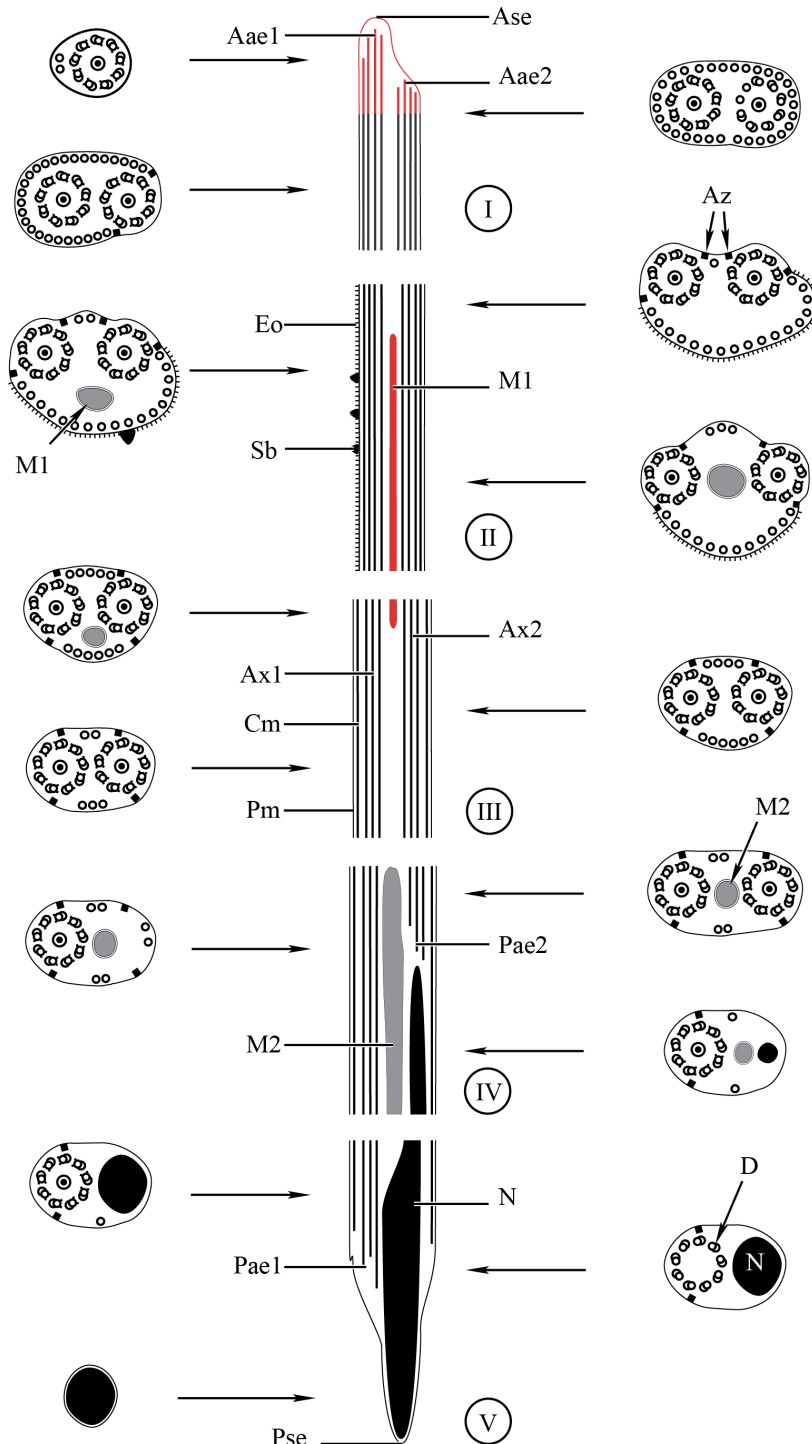

**Figure 3 Schematic reconstruction of the mature spermatozoon of *Stephanostomum murielae*.** Aae1, anterior extremity of first axoneme; Aae2, anterior extremity of second axoneme; Ase, anterior sperma-tozoon extremity; Ax1, first axoneme; Ax2, second axoneme; Cm, cortical microtubules; D, doublet; Eo, external ornamentation of the plasma membrane; M1, first mitochondrion; M2, second mitochondrion; N, nucleus; Pae1, posterior extremity of axoneme 1; Pae2, posterior extremity of axoneme 2; Pm, plasma membrane; Pse, posterior spermatozoon extremity; Sb, spine-like body.

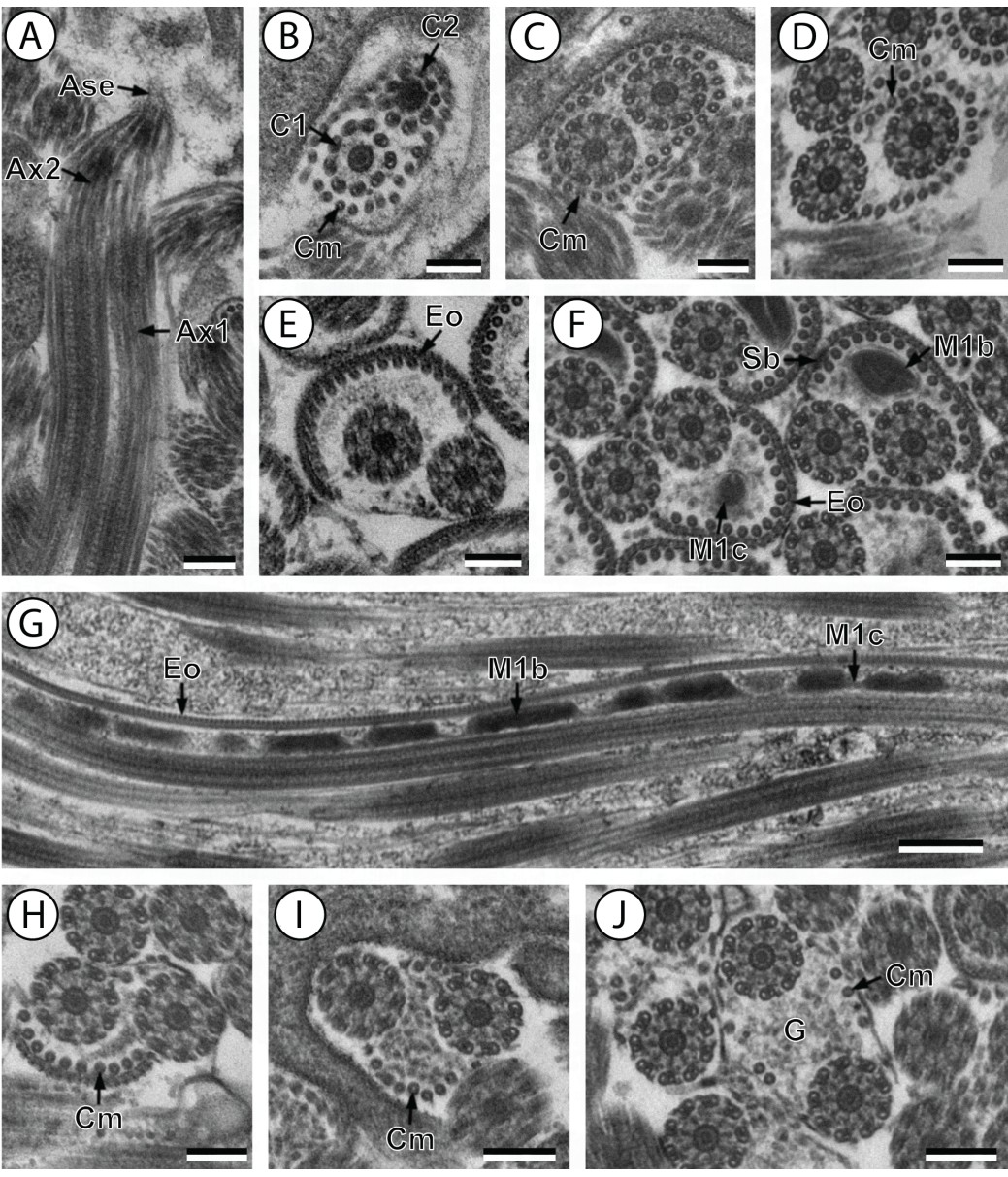

**Figure 4 Transmission electron micrographs of mature spermatozoon of *Stephanostomoides tenuis* in region I–III.** (A and B) longitudinal and cross-section sections of region I showing the anterior extremity of both axonemes 1 and 2; (C and D) consecutive cross-sections showing (C) both second axonemes surrounded by continuous layer of 31 cortical microtubules and (D) both axonemes with discontinuous layer of 23 cortical microtubules; (E–F) consecutives cross-sections of region II exhibiting the external ornamentation of the plasma membrane (E) accompanied by the first mitochondrion (moniliform) and spine-like bodies (F); (G) longitudinal section of region II, showing the moniliform mitochondrion constituted by mitochondrial bulges and mitochondrial cords; (H) distal part of region II showing external ornamentations and cortical microtubules of which maximum number is about 11; (I and J) region III or transitional area mitochondrion nor nucleus; Scale bars: 0.2 μm (A–F and H–J), 0.5 μm (g). Ase, anterior spermatozoon extremity; Ax1, first axoneme; Ax2, second axoneme; C1, centriole of the first axoneme; C2, centriole of the second axoneme; Cm, cortical microtubules; Eo, external ornamentation of the plasma membrane; G, granules of glycogen; M1b, first mitochondrion bulge; M1c, first mitochondrion cord; Sb, spine-like body.

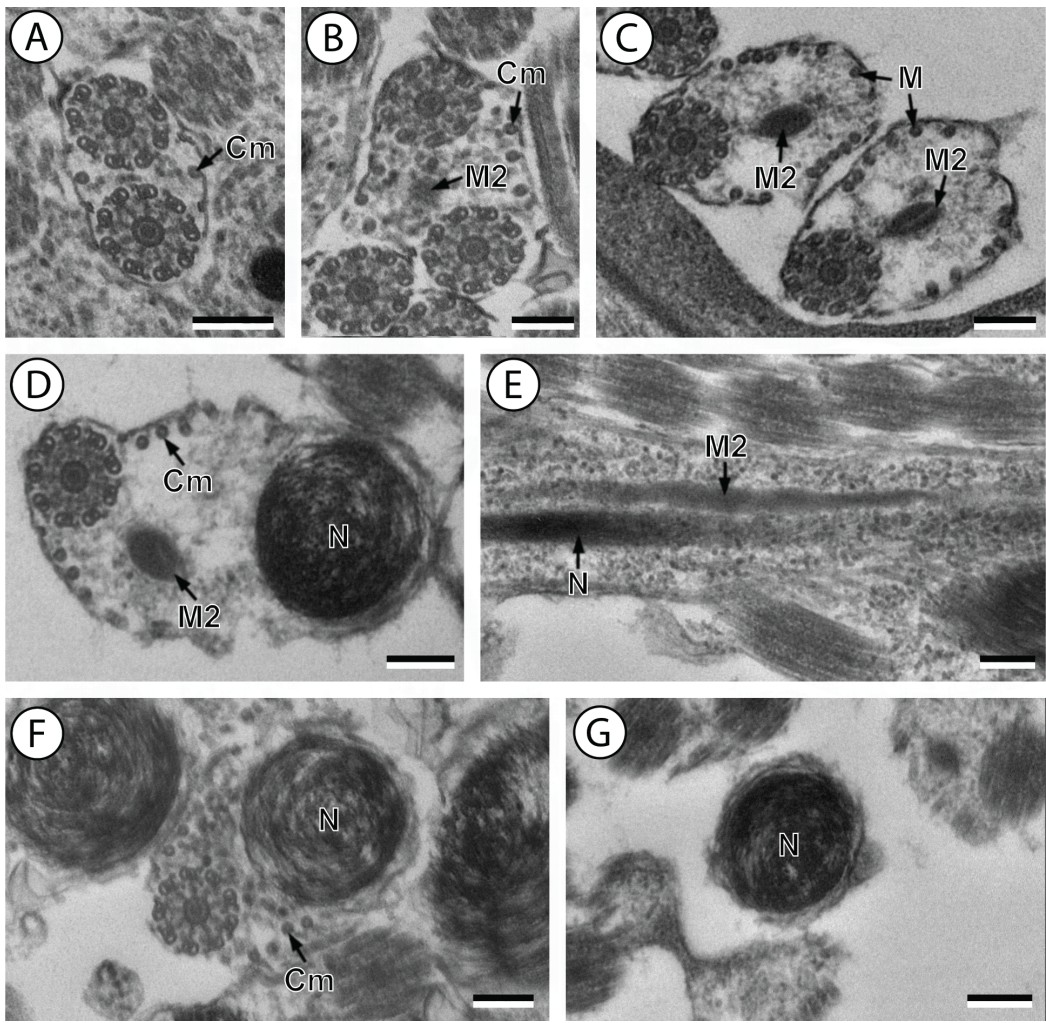

**Figure 5 Transmission electron micrographs of mature spermatozoon of *Stephanostomoides tenuis* in region III–V.** (A) cross-section in distal part of region III showing both axonemes and few microtubules (about 2); (B) proximal part of region IV showing appearance of the second mitochondrion. (C) Two cross-sections showing second mitochondrion associated with microtubules; (D) cross-section exhibiting simultaneous presence of the nucleus and the second mitochondrion accompanied by few cortical microtubules; (E) longitudinal sections showing the second mitochondrion (not moniliform); (F) cross-sections showing nucleus and few microtubules; (G) cross-sections in posterior tip of the spermatozoon exhibiting only the nucleus. Scale bars: 0.2 μm. Cm, cortical microtubules; M, microtubule; M2, second mitochondrion; N, nucleus.

is accompanied by the remaining axoneme, the second mitochondrion and some cortical microtubules (Figs. 2B, 5D, 3IV and 6IV). It is also interesting to note in *S. tenuis* the presence of a second (posterior) mitochondrion, not moniliform in longitudinal section, unlike the first (anterior) one (Figs. 5E and 6IV).

Region V is the posterior spermatozoon extremity. It is characterized in several cross-sections in *S. murielae* by the presence of the nucleus, one axoneme and few cortical microtubules that reduced progressively till their complete disappearance (Figs. 2C, 2D and 3V). In *S. tenuis*, a longitudinal section in this region V shows that the second

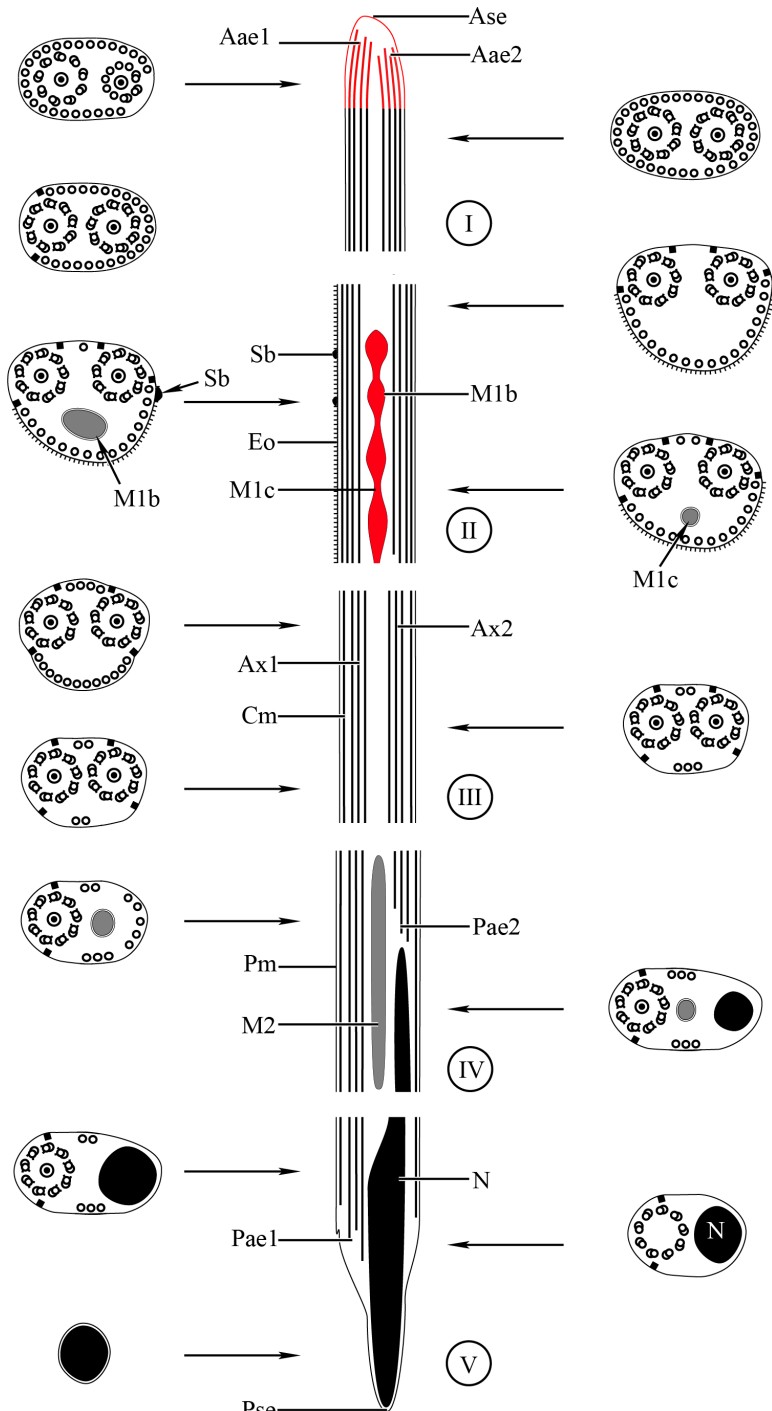

**Figure 6 Schematic reconstruction of the mature spermatozoon of *Stephanostomoides tenuis*.** Aae1, anterior extremity of first axoneme; Aae2, anterior extremity of second axoneme; Ase, anterior spermatozoon extremity; Ax1, first axoneme; Ax2, second axoneme; Cm, cortical microtubules; D, doublet; Eo, external ornamentation of the plasma membrane; M1, first mitochondrion (moniliform); M1b, first mitochondrion bulge; M1c, first mitochondrion cord; M2, second mitochondrion; N, nucleus; Pae1, posterior extremity of axoneme 1; Pae2, posterior extremity of axoneme 2; Pm, plasma membrane; Pse, posterior spermatozoon extremity; Sb, spine-like body.

mitochondrion is not moniliform (Figs. 5E and 6V). Moreover, cortical microtubules are also reduced progressively till their complete disappearance (Figs. 5F, 5G and 6V). Cross-sections in the posterior spermatozoon tip exhibit only the nucleus in both *S. murielae* and *S. tenuis* (Figs. 2E, 5G, 3V and 6V), cortical microtubules and the second axoneme having disappeared.

## DISCUSSION

The mature spermatozoa of the acanthocolpids *Stephanostomum murielae* and *Stephanostomoides tenuis* possess two axonemes of different length, showing the 9 + '1' trepaxonematan pattern, four attachment zones, two mitochondria in both species with a moniliform anterior one in *S. tenuis*, a nucleus, two bundles of parallel cortical microtubules, which maximum number in the anterior spermatozoon region is about 30 in *S. murielae* and 32 in *S. tenuis*, external ornamentation of the plasma membrane, spine-like bodies and granules of glycogen. Most of these ultrastructural characters were described previously in digeneans except in spermatozoa belonging to species of schistosomes and some didymozoids (*Justine, Jamieson & Southgate, 1993*; *Justine & Mattei, 1983*). From the anterior to the posterior extremities of the spermatozoon of both *S. murielae* and *S. tenuis*, four ultrastructural characters are compared and discussed, particularly with the other species of Lepocreadioidea.

### Anterior spermatozoon extremity

The anterior spermatozoon extremity of *Stephanostomum murielae* contains one axoneme and some cortical microtubules (about one or two) whereas in the anterior spermatozoon extremity of *Stephanostomoides tenuis* two axonemes slightly longitudinally displaced one to another were observed, surrounded by about 20 cortical microtubules. Spermatozoa showing two axonemes in their anterior extremities have also been reported in *Deropristis inflata* (*Foata, Quilichini & Marchand, 2007*) and *Neoapocreadium chabaudi* (*Kacem et al., 2010*). However, in the latter, the anterior spermatozoon extremity also contains external ornamentation of the plasma membrane. In the lepocreadioidean species studied so far, one axoneme accompanied by electron dense material in the anterior spermatozoon extremities was described in the Aephnidiogenidae *Holorchis micracanthum* (*Bâ et al., 2011*), the Gyliauchenidae *Gyliauchen* sp. and *Robphildollfusium fractum* (*Quilichini et al., 2011*; *Bakhoum et al., 2012*) and the Lepocreadiidae *Hypocreadium caputvadum* and *Opechona bacillaris* (*Kacem et al., 2012*; *Ndiaye et al., 2015*).

The distinction in the anterior spermatozoon tip of the two acanthocolpids studied here, compared to lepocreadioidean species, is the presence of cortical microtubules in the anterior tip (Table 1). In fact, some cortical microtubules are observed in the anterior spermatozoon tip in *S. murielae* (about 2) and *S. tenuis* (about 20), whereas in all the lepocreadioids species described until now the cortical microtubules are absent in the anterior spermatozoon tip. Nevertheless, their appearance is noted only when both axonemes are already formed (see *Bâ et al., 2011*; *Quilichini et al., 2011*; *Bakhoum et al., 2012*; *Kacem et al., 2012*; *Ndiaye et al., 2015*).

**Table 1** Spermatological features in Lepocreadioidea *sensu Bray (2005b)*.

| | Ase | Aldm | Eo | Eo + Cm | Teo | Sb | M | Tpse | Psc | References |
|---|---|---|---|---|---|---|---|---|---|---|
| **Apocreadiidae[a]** | | | | | | | | | | |
| *Neoapocreadium chabaudi* | Cm-Eo | − | + | + | 1 | + | 2 | ? | N? | *Kacem et al. (2010)* |
| **Acanthocolpidae[a]** | | | | | | | | | | |
| *Stephanostomum murielae* | 1Ax-Cm | − | + | + | 2 | | 2 | 2 | N | Present study |
| *Stephanostomoides tenuis* | 2Ax-Cm | − | + | + | 2 | | 2[b] | 2 | N | Present study |
| **Deropristidae[a]** | | | | | | | | | | |
| *Deropristis inflata* | 2Ax-Cm | − | + | + | 2 | − | 2 | 3? | 1Ax | *Foata, Quilichini & Marchand (2007)* |
| **Aephnidiogenidae** | | | | | | | | | | |
| *Holorchis micracanthum* | 1Ax | + | + | + | 2 | − | 1[b] | 3 | 1Ax | *Bâ et al. (2011)* |
| **Gyliauchenidae** | | | | | | | | | | |
| *Gyliauchen* sp. | 1Ax | + | + | + | 2 | + | 1 | 3 | 1Ax | *Quilichini et al. (2011)* |
| *Robphildollfusium fractum* | 1Ax | + | + | + | 2 | + | 2 | 2 | N | *Bakhoum et al. (2012)* |
| **Lepocreadiidae** | | | | | | | | | | |
| *Hypocreadium caputvadum* | 1Ax? | + | + | + | 2 | − | 2 | 3 | 1Ax | *Kacem et al. (2012)* |
| *Opechona bacillaris* | 1Ax | + | + | + | 2 | + | 2[b] | 3 | 1Ax | *Ndiaye et al. (2015)* |

**Notes.**

Ase, anterior spermatozoon extremity; Ax, axoneme; Cm, Cortical microtubules; Cm + Eo, association "cortical microtubules + external ornamentation"; Aldm, antero-lateral electron-dense material; M, number of mitochondria; N, nucleus; Psc, posterior spermatozoon character; Teo, type of external ornamentation location according to *Quilichini et al. (2011)*; Tpse, type of posterior spermatozoon extremity according *Quilichini et al. (2010b)*; Sb, spine-like bodies; +/-, presence/absence of considered character.

[a] Family excluded from the Lepocreadioidea according *Bray & Cribb*'s (*2012*) reorganization.

[b] Presence of a moniliform mitochondrion in the mature spermatozoon? Some missing micrographs not allow us to be categorical.

The variability of the anterior spermatozoon extremity may be an important criterion for phylogenetic analysis and would also be interesting when establishing spermatozoa models in digeneans.

## Antero-lateral electron-dense material

Electron dense material is located in the anterior spermatozoon extremity partially surrounding the second axoneme beneath the plasma membrane. In fact, the simultaneous presence of this antero-lateral electron-dense material and the absence of cortical microtubules in the anterior tip of the mature spermatozoon are described in some Lepocreadioidea studied (see *Ndiaye et al., 2015*, Table 1). In agreement with *Ndiaye et al. (2015)*, these features may characterize the Lepocreadioidea, distinguishing their mature spermatozoa from those of other species of digeneans. Moreover, this anterolateral electron dense material could be an apomorphic character considering its absence in most digenean spermatozoa.

In the mature spermatozoa of *S. murielae* and *S. tenuis* antero-lateral electron-dense material is not observed. The absence of antero-lateral electron-dense material is also reported in the spermatozoon of the apocreadiid *N. chabaudi* (*Kacem et al., 2010*) and deropristid *D. inflata* (*Foata, Quilichini & Marchand, 2007*). It is interesting to remark that these three families (Acanthocolpidae, Apocreadiidae and Deropristidae) are placed

outside of the Lepocreadioidea by *Bray & Cribb (2012)* in their recent re-organisation of the superfamily. The absence of antero-lateral electron-dense material in the mature spermatozoon of the studied species belonging to Acanthocolpidae (present study), Apocreadiidae (*Kacem et al., 2012*) and Deropristidae (*Foata, Quilichini & Marchand, 2007*) are an ultrastructural argument supporting the exclusion of these families from the Lepocreadioidea (Table 1). Moreover, more spermatological studies are needed on all families, but particularly the Acanthocolpidae, Apocreadiidae and Deropristidae, considering that only one study has been carried out in each family.

## External ornamentations and spine-like bodies

The presence of an association "external ornamentation + cortical microtubules + spine-like bodies" has been observed in the anterior area of the spermatozoon of most digeneans species (*Justine & Mattei, 1983*; *Justine, Jamieson & Southgate, 1993*; *Miquel et al., 2006*; *Miquel et al., 2013*; *Quilichini, Foata & Marchand, 2007*; *Bakhoum et al., 2011b*; *Bakhoum, 2012*). In the spermatozoon of *S. murielae* and in *S. tenuis* such association is observed in the area containing the first mitochondrion. The location of external ornamentations in both acanthocolpids corresponds to the type 2 of digenean spermatozoa, according to the localization of the external ornamentation established by *Quilichini et al. (2011)* (Table 1).

In the mature spermatozoon of the deropristid *D. inflata* (*Foata, Quilichini & Marchand, 2007*), external ornamentations are present associated with cortical microtubules but spine-like bodies are absent. In the apocreadiid *N. chabaudi* (*Kacem et al., 2010*), spine-like bodies are present but not in the ornamented area of the spermatozoon.

Within the superfamily Lepocreadioidea as recognised by *Bray & Cribb (2012)* the association "external ornamentation + cortical microtubules" is present in all species studied (see *Ndiaye et al., 2015*, Table 1). However, spine-like bodies are absent in *H. micracanthum* (*Bâ et al., 2011*) and *H. caputvadum* (*Kacem et al., 2012*), while they are present in *Gyliauchen* sp. (*Quilichini et al., 2011*), *R. fractum* (*Bakhoum et al., 2012*) and *O. bacillaris* (*Ndiaye et al., 2015*).

What we refer to as 'spine-likes bodies,' are reported, to our knowledge, only in the spermatozoa of digeneans (see *Bakhoum, 2012*; *Bruňanská et al., 2014*) and consequently need more attention in phylogenetic analyses.

## Number and morphology of mitochondria

The mature spermatozoa of *Stephanostomum murielae* and *Stephanostomoides tenuis* possess two mitochondria. The first mitochondrion is located at the level of the external ornamentation of the plasma membrane, and the second one in the area containing the nucleus. In the spermatozoon of *Stephanostomoides tenuis* a particular anterior mitochondrion is observed, appearing as successive bulges interconnected by a cord. This type of mitochondrion was named "moniliform" by *Bâ et al. (2011)* and was also reported in *O. bacillaris* by *Ndiaye et al. (2015)* (Table 1). The presence of two mitochondria has been described in several species, including the apocreadiid *N. chabaudi* (*Kacem et al., 2010*), the deropristid *D. inflata* (*Foata, Quilichini & Marchand, 2007*), the gyliauchenid

*R. fractum* (*Bakhoum et al., 2012*), the lepocreadiids *H. caputvadum* (*Kacem et al., 2012*) and *O. bacillaris* (*Ndiaye et al., 2015*), the fasciolid *Fasciola hepatica* (*Ndiaye et al., 2003*), and the troglotrematid *Troglotrema acutum* (*Miquel et al., 2006*). However, in most digenean spermatozoa studied until now only one mitochondrion is reported (see *Bakhoum, 2012*).

It is also important to notice that three mitochondria were also described in the mature spermatozoon of the paramphistomids *Basidiodiscus ectorchus* and *Sandonia sudanensis* (*Ashour, Garo & Gamil, 2007*), the cryptogonimid *Anisocoelium capitellatum* (*Ternengo et al., 2009*) and the heterophyid *Euryhelmis squamula* (*Bakhoum et al., 2009*).

In the present study, the morphology of the first mitochondrion is the main ultrastructural characteristic that distinguishes the mature spermatozoa of *Stephanostomum murielae* from those of *Stephanostomoides tenuis*.

## Morphology of posterior spermatozoon extremity

The morphology of the posterior spermatozoon extremity presents a great variability that emphasise the usefulness of this criterion when establishing spermatozoon models in Digenea (*Quilichini et al., 2010b*; *Bakhoum, 2012*). In *S. murielae* and *S. tenuis* the posterior spermatozoon extremity is devoid of cortical microtubules and exhibits only the nucleus. This morphology corresponds to the fasciolidean type or type 2 according to *Quilichini et al. (2010b)*. A posterior spermatozoon extremity showing only a nucleus is reported in most digeneans (*Ndiaye et al., 2003*; *Ndiaye et al., 2012*; *Quilichini et al., 2010b*; *Bakhoum et al., 2011a*; *Bakhoum et al., 2011b*). In the superfamily Lepocreadioidea (*Bray & Cribb, 2012*), a posterior spermatozoon extremity containing only a nucleus is observed in *Robphildollfusium fractum* (*Bakhoum et al., 2012*) while in the remaining species only one axoneme is present as described in *Holorchis micracanthum* (*Bâ et al., 2011*), *Gyliauchen* sp. (*Quilichini et al., 2011*), *Hypocreadium caputvadum* (*Kacem et al., 2012*) and *Opechona bacillaris* (*Ndiaye et al., 2015*) (Table 1).

Additionally, a posterior spermatozoon extremity containing cortical microtubules has been reported in other digeneans, particularly in the families Opecoelidae and Opistholebetidae (*Miquel, Nourrisson & Marchand, 2000*; *Levron, Ternengo & Marchand, 2004*; *Quilichini et al., 2007*). Moreover, in the lecithasterid *Aponurus laguncula* (*Quilichini et al., 2010a*) the posterior spermatozoon extremity containing a mitochondrion has been described.

These different types of posterior spermatozoon extremity could be used when establishing spermatozoon models considering that this criterion may allow distinguishing families or superfamilies within the Digenea.

## Phylogenetic significances

The mature spermatozoon of *Stephanostomum murielae* and *Stephanostomoides tenuis* share several ultrastructural features (Table 1). The main differences in their mature spermatozoon are the morphology of the anterior spermatozoon extremity and that of the first mitochondrion. Thus, the spermatological similarities between the best known genus *Stephanostomum* with over 100 nominal species and the genus *Stephanostomoides* with

only 2 nominal species confirm that the later belongs to the same family Acanthocolpidae. Moreover, *Stephanostomoides* is considered as the genuine genus of acanthocolpid based on its morphological similarity to *Acanthocolpus* and *Stephanostomum* (*Bray et al., 2005*). However, spermatological evidence from the type-genus *Acanthocolpus* would improve our understanding of the ultrastructural characteristics in the Acanthocolpidae.

Molecular evidence (*lsr*DNA) of the position of the Acanthocolpidae presented by *Curran, Tkach & Overstreet (2006)* suggested that it should be placed in the superfamily Brachycladioidea, along with the Opecoelidae, Opistholebetidae and Brachycladiidae. *Bray et al. (2009)* using *lsr*DNA + *nad1* sequences presented evidence that the Acanthocolpidae and Brachycladiidae were, at least, closely related, and were distinct from the Lepocreadioidea. *Littlewood, Bray & Waeschenbach (in press)* utilised 202 *ssrDNA* and 556 *lsrDNA* trematode sequences to produce an estimate of the phylogeny of the group. These genes were found to be useful in defining the constituents of superfamilies, but much less satisfactory for assessing the relationships between superfamilies. Their results indicated that of the families placed in the Brachycladioidea by *Curran, Tkach & Overstreet (2006)* only the Brachycladiidae and Acanthocolpidae were consistently resolved together, and that the Opecoelidae and Opistholebetidae are best considered in a separate superfamily, the Opecoeloidea. Unfortunately, ultrastructural data are lacking for the Brachycladiidae to corroborate the close relationships between acanthocolpids and brachycladiids. However, spermatological characters observed in some species belonging to the Opecoelidae and Opistholebetidae reveal several differences between mature spermatozoa of opecoelids + opistholebetids and those of the two acanthocolpids described here. One of the main differences between these clades concerns the morphology of the posterior spermatozoon extremity as stated above. These spermatological differences corroborate the molecular findings that the Acanthocolpidae is not close to the clade Opecoelidae + Opistholebetidae.

### Funding

AJS Bakhoum is funded by a post-doctoral fellowship: no CE/01/2013 from the Collectivité Territoriale de Corse—Direction de l'Enseignement Supérieur et de la Recherche. The funders had no role in study design, data collection and analysis, decision to publish, or preparation of the manuscript.

### Grant Disclosures

The following grant information was disclosed by the authors:
Collectivité Territoriale de Corse—Direction de l'Enseignement Supérieur et de la Recherche: CE/01/2013.

### Competing Interests

Jean-Lou Justine is an Academic Editor for PeerJ.

## Author Contributions

- Abdoulaye J.S. Bakhoum conceived and designed the experiments, performed the experiments, analyzed the data, wrote the paper.
- Yann Quilichini and Bernard Marchand conceived and designed the experiments, performed the experiments, analyzed the data, contributed reagents/materials/analysis tools, prepared figures and/or tables, reviewed drafts of the paper.
- Jean-Lou Justine conceived and designed the experiments, performed the experiments, contributed reagents/materials/analysis tools, reviewed drafts of the paper.
- Rodney A. Bray performed the experiments, contributed reagents/materials/analysis tools, reviewed drafts of the paper.
- Cheikh T. Bâ reviewed drafts of the paper.

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
