# Peer review of "Ultrastructural study of sperm cells in Acanthocolpidae: the case of Stephanostomum murielae and Stephanostomoides tenuis (Digenea)"

_PeerJ, doi:10.7717/peerj.744_

## Round 0.1 · original submission · Minor Revisions

Please, make the reviewer's changes as suggested.

Reviewer 1 ·

Basic reporting

No Comments

Experimental design

No Comments

Validity of the findings

No Comments

Additional comments

The paper describes the ultrastructure of the spermatozoon of the Digeneans Stephanostomum murielae and Stephanostomoides tenuis; the methods applied are appropriate and the micrographs presented are of good quality. This contribution shows novel, interesting results, but does not provide enough discussion in relation to the phylogenetic aspects. In addition, ultrastructural characters are of no phylogenetic value as long as they do not follow the procedere of testing phylogenetic relationships (statement of plesiomorphic and apomorphic conditions of a character, outgroup comparison etc.). And here lies a substantial deficiency of the paper: The phylogenetic considerations are no consequent and compelling. At the end of the Discussion, there are some suggested phylogenetical implications, assumed on the basis of the differences revealed in the spermatozoa ultrastructure, and combined with molecular findings. However, more discussion on this topic is needed, otherwise the title does not reflect the content of the paper. The authors are undoubtedly excellent specialists on digenean ultrastructure and systematic but the phylogenetic aspects of this paper are not strong.
More specific comments:
line 222: please, update citation with the most recent paper on the fine structure of digenean spermatozoon (e.g. Parasitology Research (2014) 113: 2483-2491)
line 257: how many nominated species includes the genus Stephanostomoides?
line 252 and paragraph Phylogenetical significances: please, analyse in more detail ultrastructural features of the digenean spermatozoa and their utilisation as phylogenetic characters for the Digenea.

Reviewer 2 ·

Basic reporting

No Comments

Experimental design

No Comments

Validity of the findings

No Comments

Additional comments

This is another good contribution to the use of parasite reproduction characters to phylogenetic inferences. I think this ms should be accepted after only some minor modifications that should be made to the ms.
line 27 – “this clade at the family rank” should be changed to “clade to the family rank”; “Since, this family” should be changed to “Since then, this family”
line 29 “are included” should be changed to “were included”
line 76 units for the mesh size are missing.
line 49 there is a parenthesis missing
line 78 “section placed” should be changed to “sections placed”
line 90 “surrounding” should be changed to “surrounded”
line 92 continuous
line 95 “into” should be changed to “to”
lines 108-109 parallel bundles of cortical MTs or “bundles of parallel cortical MTs”?
line 110 In S. tenuis the region…
line 117”… one of the axonemes”
line 136 “bundles of parallel cortical microtubules, which…”
line 139 use “were described”
line 151 use “ the latter”
line 172 use “In agreement with…”
line 176-178 too confusing: please rewrite.
line 249 use “could” instead of “would”

---

## Round 0.2 · Minor Revisions

Authors should carefully checked the revised manuscript because the tittle is missing and because the changes made in the text do not correspond to the changes indicated in the rebuttal letter.

---

## Round 0.3 · accepted · Accept

You have considered all the comments and suggestions made by the reviewers in the new version of your manuscript which has been improved as a result.